# The *Plasmodium falciparum* rhoptry protein RhopH3 plays essential roles in host cell invasion and nutrient uptake

**Emma S Sherling[1,2], Ellen Knuepfer[3], Joseph A Brzostowski[4], Louis H Miller[2], Michael J Blackman[1,5]\*, Christiaan van Ooij[1]\***

[1]Malaria Biochemistry Laboratory, The Francis Crick Institute, London, United Kingdom; [2]Laboratory of Malaria and Vector Research, National Institute of Allergy and Infectious Diseases, National Institutes of Health, Rockville, United States; [3]Malaria Parasitology Laboratory, The Francis Crick Institute, London, United Kingdom; [4]Laboratory of Immunogenetics Imaging Facility, National Institute of Allergy and Infectious Diseases, National Institutes of Health, Rockville, United States; [5]Department of Pathogen Molecular Biology, London School of Hygiene & Tropical Medicine, London, United Kingdom

**\*For correspondence:** Mike. Blackman@crick.ac.uk (MJB); Christiaan.vanOoij@crick.ac.uk (CvO)

**Competing interests:** The authors declare that no competing interests exist.

**Abstract** Merozoites of the protozoan parasite responsible for the most virulent form of malaria, *Plasmodium falciparum,* invade erythrocytes. Invasion involves discharge of rhoptries, specialized secretory organelles. Once intracellular, parasites induce increased nutrient uptake by generating new permeability pathways (NPP) including a *Plasmodium* surface anion channel (PSAC). RhopH1/Clag3, one member of the three-protein RhopH complex, is important for PSAC/NPP activity. However, the roles of the other members of the RhopH complex in PSAC/NPP establishment are unknown and it is unclear whether any of the RhopH proteins play a role in invasion. Here we demonstrate that RhopH3, the smallest component of the complex, is essential for parasite survival. Conditional truncation of RhopH3 substantially reduces invasive capacity. Those mutant parasites that do invade are defective in nutrient import and die. Our results identify a dual role for RhopH3 that links erythrocyte invasion to formation of the PSAC/NPP essential for parasite survival within host erythrocytes.

## Introduction

Parasites of the genus *Plasmodium* are the causative agents of malaria, a disease that claims nearly 600,000 lives each year (**WHO, 2014**). Of the five *Plasmodium* species that infect humans, *Plasmodium falciparum* is responsible for nearly all the mortality associated with malaria. The disease is the result of asexual replication of the parasite in erythrocytes. For approximately the first half of the 48 hour *P. falciparum* intraerythrocytic life cycle, the parasite exists in a mononuclear trophozoite form (the earliest stages of which are generally referred to as ring stages), during which the parasite grows rapidly. During this phase, *P. falciparum*-infected erythrocytes gain the capacity to adhere to host vascular endothelium, a process that depends on the export of parasite proteins to form adhesive structures called knobs at the host erythrocyte surface. Nuclear division then commences, initiating differentiation into a schizont (a process called schizogony). This multinucleated form eventually undergoes segmentation to form invasive merozoites that egress upon rupture of the infected erythrocyte to invade new erythrocytes.

Egress and erythrocyte invasion involves the regulated discharge of several sets of apical merozoite secretory organelles that are unique to apicomplexan parasites. The largest of these organelles,

**eLife digest** Malaria is a life-threatening disease that affects millions of people around the world. The parasites that cause malaria have a complex life cycle that involves infecting both mosquitoes and mammals, including humans. In humans, the parasites spend part of their life cycle inside red blood cells, which causes the symptoms of the disease. In order to survive and multiply, malaria parasites need to make the red blood cell more permeable so that it can absorb nutrients from the blood stream and get rid of the toxic waste products they generate.

It remains unclear how the parasites do this, but previous research has shown that the parasites produce channel-like proteins that make red blood cells more permeable to nutrients. One of the proteins involved in this process forms part of a complex with two other proteins, called RhopH2 and RhopH3. It is not known what these other two proteins do, and whether they are necessary for creating the new nutrient channels.

Sherling et al. studied the RhopH3 protein to see if it is required to make red blood cells more permeable. The experiments used a genetically modified version of the parasite, in which RhopH3 no longer interacted with the two other proteins. The findings show that RhopH3 has two important roles: first, parasites need it to invade the red blood cells, and second, parasites cannot get nutrients into the red blood cell without RhopH3.

Most antimalarial drugs work by preventing parasite replication in red blood cells, but parasites are becoming increasingly resistant to these drugs. Understanding which proteins allow parasites to invade and grow within blood cells will further the development of new malaria medication. The next step will be to understand the molecular mechanisms by which RhopH3 promotes invasion and subsequently facilitates nutrient uptake, and will help researchers to explore its potential as a drug target.

called rhoptries, contain several proteins involved in adhesion to the host cell. Rhoptries are also thought to mediate formation of the nascent parasitophorous vacuole (PV), a membranous compartment that surrounds the parasite after entry has been completed (*Carruthers and Sibley, 1997*; *Counihan et al., 2013*). Despite the importance of rhoptries in invasion and subsequent host cell remodeling, a detailed understanding of the function of many rhoptry proteins is lacking. Rhoptries comprise at least two subdomains (*Counihan et al., 2013*) referred to as the rhoptry neck and the rhoptry bulb. The contents of these subdomains likely mediate different functions, as reflected by evidence suggesting that they are released sequentially during invasion (*Zuccala et al., 2012*). Proteins of the rhoptry neck are well conserved between *Plasmodium spp.* and the related apicomplexan parasite *Toxoplasma gondii*, suggesting conserved functions (*Counihan et al., 2013*; *Proellocks et al., 2010*). In contrast, proteins of the rhoptry bulb appear to be genus-specific, perhaps reflecting functions unique to each parasite (*Counihan et al., 2013*). A function for several *P. falciparum* rhoptry bulb proteins has been proposed, such as a role for a protein called RAMA in transport of proteins to the rhoptry (*Richard et al., 2009*), but the inability to produce mutants lacking these proteins has precluded conclusive assignments of function (*Kats et al., 2006*). Hence, the molecular functions of most rhoptry proteins remain unknown.

One component of the *P. falciparum* rhoptry bulb that has received particular attention is the so-called high molecular weight (HMW) rhoptry or RhopH complex, which consists of three proteins called RhopH1/Clag, RhopH2, and RhopH3 (*Cooper et al., 1988*; *Holder et al., 1985a*). Whilst RhopH2 and RhopH3 are each encoded by single-copy genes, RhopH1/Clag, the largest component of the complex, exists in five isotypes encoded by separate genes entitled *clag2*, *clag3.1*, *clag3.2*, *clag8* and *clag9* (*Kaneko et al., 2001*, *2005*). RhopH1/Clag3.1 and RhopH1/Clag3.2 are nearly identical proteins that are expressed in a mutually exclusive manner (*Chung et al., 2007*; *Comeaux et al., 2011*; *Cortés et al., 2007*). Each RhopH complex contains only one form of RhopH1/Clag (*Kaneko et al., 2005*), so each parasite has the potential to produce four different RhopH complexes, differentiated by the particular RhopH1/Clag isotype bound. All members of the RhopH complex are expressed late in the intraerythrocytic cycle (*Cooper et al., 1988*). The complex

is then released during invasion (*Ling et al., 2003*) and inserted into the nascent PV membrane (PVM) soon after parasite entry (*Ling et al., 2004*; *Sam-Yellowe et al., 1988*).

Genetic and chemical genetic investigation has revealed a role for the RhopH1/Clag3 proteins in the function of the *Plasmodium* surface anion channel (PSAC), a new permeability pathway (NPP) induced in host erythrocytes following parasite entry and involved in nutrient acquisition by the intracellular parasite (*Nguitragool et al., 2011*). Pharmacological inhibition of RhopH1/Clag3.2 function was found to block PSAC/NPP activity, and selection for drug-resistant mutants revealed that part of the protein is exposed at the surface of the erythrocyte and that it may form the channel itself (*Nguitragool et al., 2014*; *Sharma et al., 2015*). However, parasites that do not produce either RhopH1/Clag3.1 or RhopH1/Clag3.2 display only a small growth disadvantage (*Comeaux et al., 2011*) and inhibition of the function of these proteins has only a small effect on parasite growth rates *in vitro* (*Pillai et al., 2012*). Parasites lacking RhopH1/Clag9 are viable, and an early report suggested that loss of the *clag9* gene resulted in loss of binding to CD36 (*Trenholme et al., 2000*). However, this has been disputed (*Nacer et al., 2011*), as a subsequent study identified another gene closely linked to the *clag9* gene that is important for CD36 binding (*Nacer et al., 2015*). Hence, whilst the function of RhopH1/Clag9 remains to be determined, like RhopH1/Clag3.1 and RhopH1/Clag 3.2, it is not essential. There are no reports describing a deletion, or attempted deletion, of *clag2* or *clag8*.

Much less is known of the function of the RhopH2 and RhopH3 components of the complex. There is no report of attempted disruption of the *rhopH2* gene, but the *rhopH3* gene is refractory to deletion in the haploid blood stages (*Cowman et al., 2000*), suggesting an essential role. Hints that this might include a function in invasion derive from studies showing that antibodies to RhopH3 can inhibit invasion (*Cooper et al., 1988*; *Doury et al., 1994*). However, whether RhopH3 plays other essential roles that involve all forms of the RhopH complex is unknown.

Here we use a conditional mutagenesis approach to modify the *rhopH3* gene in a manner that prevents formation of the RhopH complex. The resulting mutant parasites show two distinct phenotypes: a significant decrease in the level of erythrocyte invasion and a complete block in intracellular development at the trophozoite stage. Our findings reveal that RhopH3 and the RhopH complex have essential roles in two distinct stages of the erythrocytic lifecycle.

## Results

### Efficient conditional truncation of the *rhopH3* gene

Previous attempts to disrupt the *P. falciparum rhopH3* gene using conventional genetic techniques were unsuccessful (*Cowman et al., 2000*), suggesting an indispensable role in asexual blood stages. To gain insights into this role we therefore adopted the DiCre conditional recombinase system recently adapted to *P. falciparum* (*Collins et al., 2013*) to examine the consequences of functional inactivation of RhopH3. For this, we used Cas9-mediated genome editing (*Ghorbal et al., 2014*) to introduce synthetic introns containing *loxP* sites (*Jones et al., 2016*) into the *rhopH3* gene such that they flanked an internal region spanning exons 4–6, the region of the gene that shows the highest level of conservation across *Plasmodium rhopH3* orthologs (*Figure 1A*, *Figure 1—figure supplement 1*). This genomic modification was made in the DiCre-expressing *P. falciparum* 1G5DC parasite clone (*Collins et al., 2013*) in order that excision of the floxed sequence could be induced by treatment of the transgenic parasites with rapamycin. DiCre-mediated excision was predicted to generate an internally-truncated mutant form of *rhopH3* lacking its most highly conserved region.

Successful modification of the *rhopH3* gene in the transfected parasite population following introduction of the targeting vector was confirmed by diagnostic PCR (not shown). Subsequent limiting dilution cloning of the modified parasites resulted in the isolation of parasite clones *rhopH3-loxP* 5F5 and *rhopH3-loxP* 4B11, which were derived from independent transfections using different guide RNAs. Modification of the native *rhopH3* locus in the expected fashion was confirmed in both parasite clones by diagnostic PCR and Southern blot (*Figure 1B* and *Figure 1C*). Both clones displayed RhopH3 expression levels and *in vitro* replication rates indistinguishable from the parental 1G5DC parasites (*Figure 1—figure supplement 2*), indicating that the modified *rhopH3* gene generated wild type levels of RhopH3 and that the modifications had no impact on parasite viability. The clones were therefore used for all subsequent experiments.

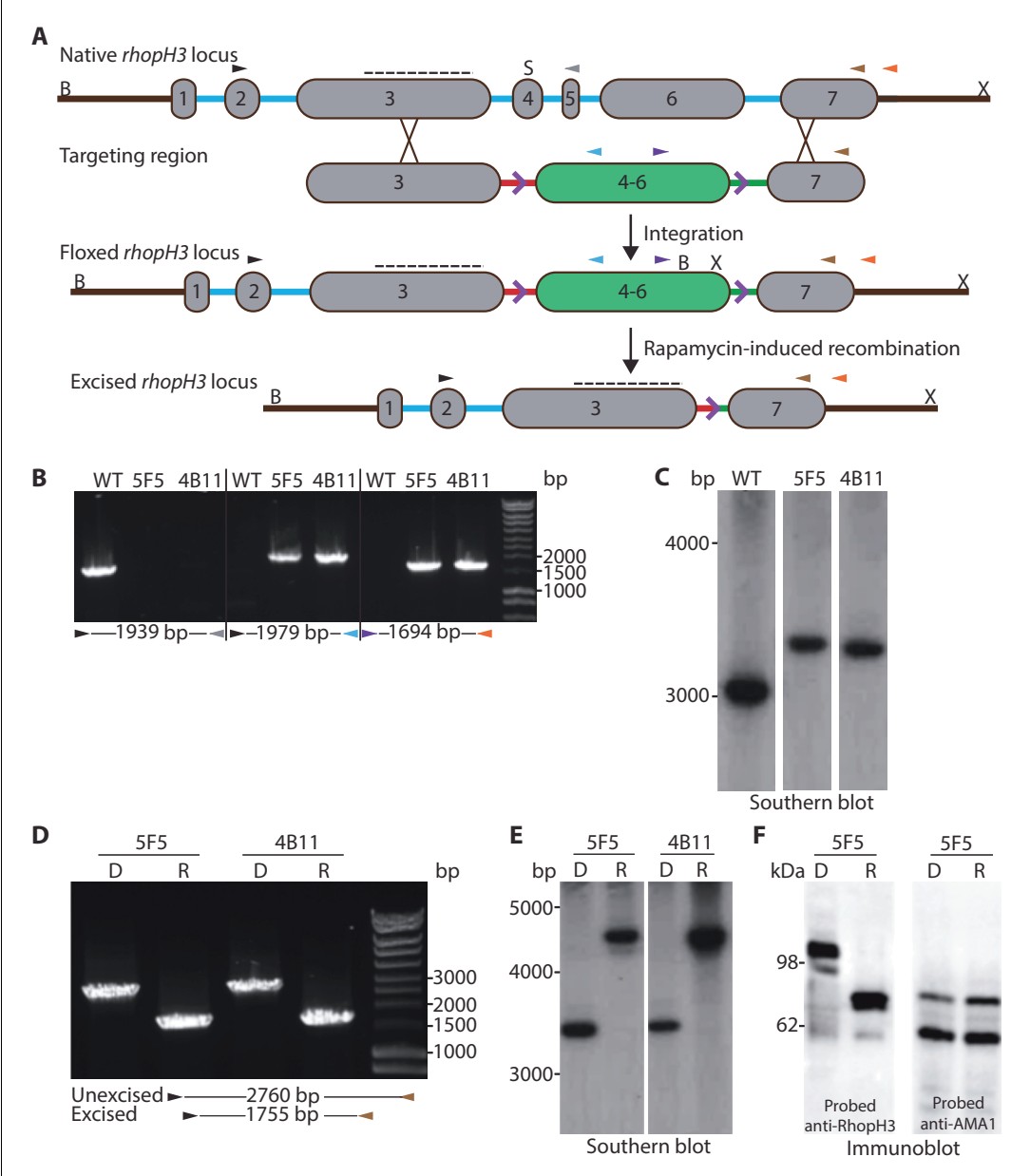

**Figure 1.** Conditional truncation of the *rhopH3* gene. (**A**) The *rhopH3* gene comprises seven exons (numbered grey boxes) and six introns (blue lines). Using Cas9-mediated recombination, the region spanning introns 3 to 6 was replaced with two *loxP*-containing (purple open arrowhead) *P. falciparum* introns (*SERA2* (orange line) and *sub2* (green line)) flanking a recodonized and fused version of exons 4 to 6 (exon 4–6, green box). Integration of this sequence by homologous recombination was promoted by the addition of sequences of exon 3 and 7 to either side of the introns. Colored arrowheads, primer binding sites. B, S and X, BsgI, SacI and XmnI restriction sites. Dotted line, probe used for Southern blotting. Rapamycin-induced site-specific recombination between the *loxP* sites removes the recodonized exon 4–6. (**B**) PCR analysis of *rhopH3-loxP* clones 5F5 and 4B11 confirms the expected gene modification event. Genomic DNA from parental 1G5DC (WT) parasites or the clones was used as template for PCR using the indicated primers (see panel A). Numbers between the arrowheads indicate the expected size of the amplicon. (**C**) Southern blot analysis of parental 1G5DC (WT) and the *rhopH3-loxP* parasite clones confirms the expected modification of the *rhopH3* locus. Genomic DNA was digested with BsgI, SacI and XmnI and hybridized with a radiolabeled probe that binds to part of exon 3 (dotted line in panel A). Expected fragment sizes are 3016 bp for the WT *rhopH3* locus and 3349 bp for the *rhopH3-loxP* locus. (**D**) Efficient rapamycin-induced truncation of the *rhopH3* gene. Clones *rhopH3-loxP* 5F5 and 4B11 were analyzed by PCR ~44 hr after treatment with DMSO (D) or rapamycin (R) using the indicated primers (see panel A). Excision decreases the amplicon from 2760 bp to 1755 bp. (**E**) Southern blot showing efficient rapamycin-induced truncation of the *rhopH3* gene. Genomic DNA extracted from control or rapamycin-treated *rhopH3-loxP* clones 5F5 and 4B11 was digested and probed as described in panel C. Expected fragment sizes are 3349 bp for the non-excised locus and 4784 bp for the excised locus. (**F**) Immunoblot analysis of mature schizonts of *rhopH3-loxP* clone 5F5, examined ~44 hr following treatment at ring stage with DMSO (D) or rapamycin (R). The blots were probed

*Figure 1 continued on next page*

*Figure 1 continued*

with an antibody against RhopH3 (left panel) or the merozoite protein AMA1 (right panel) as a loading control. The expected molecular masses of WT RhopH3 and RhopH3△4–6are ~110 kDa and~70 kDa, respectively. In panels B–F, positions of relevant molecular mass markers are indicated.

The following figure supplements are available for figure 1:

**Figure supplement 1.** Multiple alignment of predicted primary sequences of *rhopH3* orthologues from *P. falciparum* (PF3D7_0905400), *Plasmodium chabaudi* (PCHAS_0416900) and *Plasmodium vivax* (PVX_098712).

**Figure supplement 2.** Modification (floxing) of the *rhoph3* gene does not impact on gene expression or parasite growth.

**Figure supplement 3.** Conditional truncation of RhopH3 in both the 5F5 and 4B11 *rhopH3-loxP* clones.

To examine the efficiency of conditional excision of the floxed sequence in the *rhopH3-loxP* clones, tightly synchronized ring stage cultures of both clones were divided into two and treated for 4 hr with either rapamycin or DMSO (vehicle control). Following washing and further incubation for ~44 hr to allow maturation of the parasites to schizont stage (at which peak expression of RhopH3 occurs (*Cooper et al., 1988*), genomic DNA from the clones was examined by PCR and Southern blot. This revealed highly efficient excision of the floxed *rhopH3* sequence (*Figure 1D* and *Figure 1E*).

DiCre-mediated site-specific recombination between the introduced *loxP* sites in the modified *rhopH3* locus of the *rhopH3-loxP* parasites was expected to reconstitute a functional, albeit chimeric, intron. Upon splicing of this intron exons 3 and 7 are placed in frame, producing a modified RhopH3 gene product (called RhopH3△4–6) that retains wild type N-terminal and C-terminal segments but lacks residues encoded by exons 4–6. Extracts of the rapamycin-treated and control parasites were analyzed by immunoblot ~44 hr following treatment using antibody anti-Ag-44, which recognizes an epitope within the C-terminal segment of RhopH3 encoded by exon 7 (*Coppel et al., 1987*). As shown in *Figure 1F* and *Figure 1—figure supplement 3*, rapamycin treatment produced the expected change in mass, converting the ~110 kDa wild type RhopH3 to a ~70 kDa RhopH3△4–6 product. This conversion was highly efficient, with no residual full-length protein detected in the rapamycin-treated schizonts. These results confirmed the excision data and demonstrated essentially complete conditional truncation of RhopH3 within a single erythrocytic cycle in the *rhopH3-loxP* parasite clones.

## Truncation of *rhopH3* leads to mislocalization of other components of the RhopH complex

We next aimed to determine the effects of RhopH3 truncation on its subcellular localization within the parasite, as well as on the trafficking of other members of the RhopH complex. Immunofluorescence analysis (IFA) showed that, as expected, RhopH3 colocalized with the rhoptry marker RAP2 (*Bushell et al., 1998*; *Crewther et al., 1990*) in mature schizonts of control *rhopH3-loxP* parasites (*Figure 2A*). However, in rapamycin-treated (RhopH3△4–6) parasites, this colocalization was lost, although RAP2 was still detected in a punctate, apically-disposed pattern typical of rhoptries (*Figure 2A*). To determine the effects of this mistrafficking on localization of the other two RhopH complex proteins, control and rapamycin-treated *rhopH3-loxP* parasites were probed with anti-RAP2 as well as either anti-RhopH1/Clag3.1 (*Kaneko et al., 2005*) or anti-RhopH2 antibodies (*Holder et al., 1985a*). This showed that, as in the case of RhopH3, rhoptry localization of both RhopH1/Clag3.1 and RhopH2 was lost in rapamycin-treated parasites (*Figure 2A*). These results indicated that truncation of RhopH3 to the RhopH3△4–6 form resulted in mistrafficking of at least some components of the RhopH complex. To determine whether the mistrafficked rhoptry proteins all localized to the same parasite compartment, the parasites were co-stained with various combinations of antibodies against two of the three complex proteins. This showed that neither RhopH2 nor RhopH1/Clag3.1 colocalized with RhopH3△4–6 in the mutant parasites (*Figure 2B*). The RhopH2 and RhopH1/Clag3.1 signals were also distinct in the mutant parasites, although in this case some limited colocalization of these proteins was apparent (bottom images, *Figure 2B*).

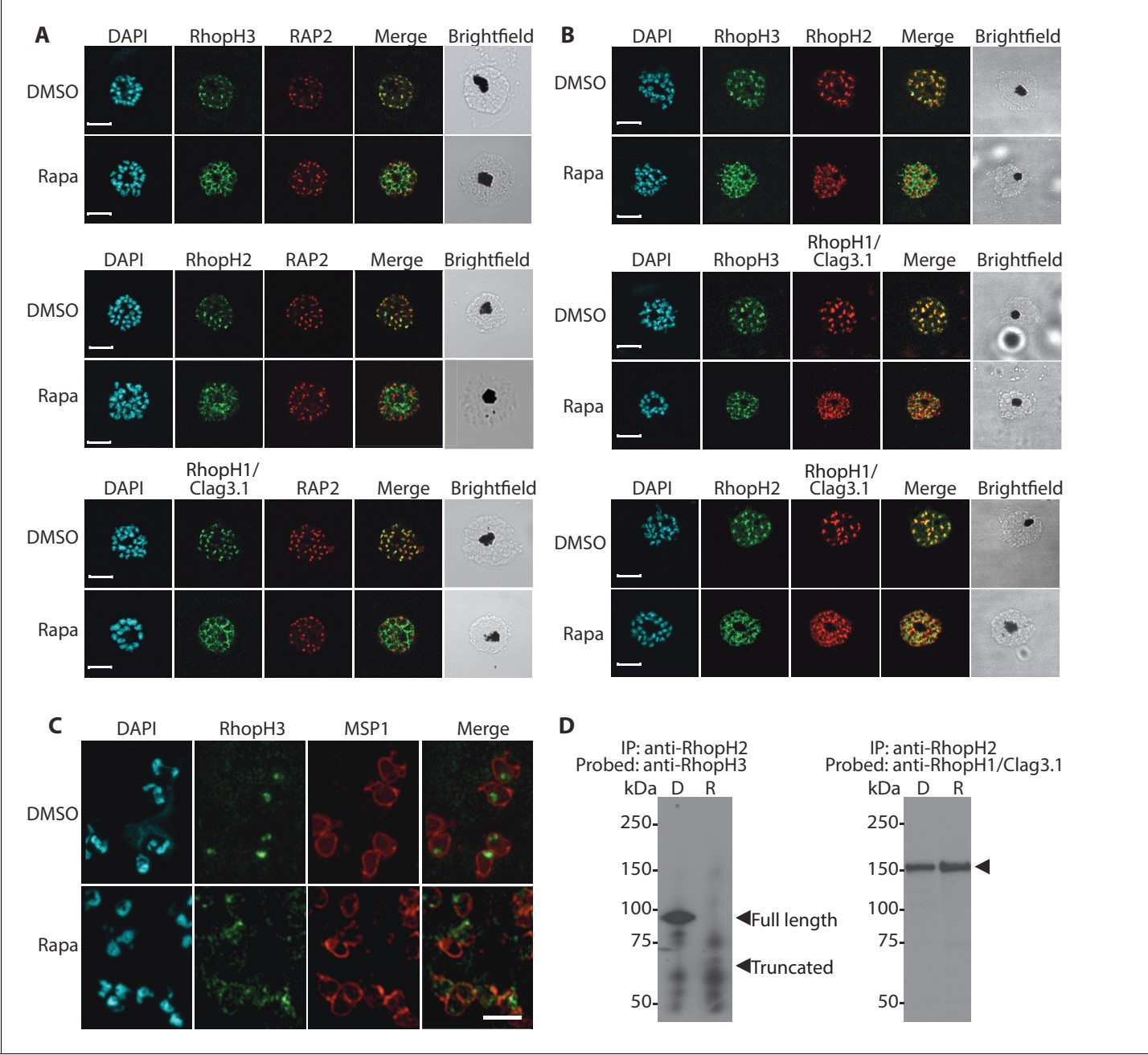

**Figure 2.** Truncation of RhopH3 leads to mistrafficking of components of the RhopH complex and loss of complex formation. (**A**) IFA showing colocalization of RhopH3, RhopH2 and RhopH1/Clag3.1 with the rhoptry marker RAP2 in schizonts of control (DMSO) *rhopH3-loxP* parasites but loss of colocalization following rapamycin (Rapa) treatment. Parasite nuclei were visualized by staining with 4,6-diamidino-2-phenylindole (DAPI). Scale bar, 5 µm. (**B**) Colocalization of the members of the RhopH complex. RhopH3, RhopH2 and RhopH1/Clag3.1 colocalize in *rhopH3-loxP* parasites treated with DMSO, but this colocalization is lost in parasites treated with rapamycin. (**C**) Mislocalisation and reduced levels of RhopH3 in naturally released free merozoites of *rhopH3-loxP* parasites treated with rapamycin. Samples were probed with a monoclonal antibody to the merozoite surface marker MSP1 as well as anti-RhopH3 antibodies. Scale bar, 2 µm. (**D**) Immunoprecipitation reveals disruption of the RhopH complex in rapamycin-treated *rhopH3-loxP* parasites. RhopH2 was immunoprecipitated from extracts of control or rapamycin-treated *rhopH3-loxP* parasites. Subsequent immunoblotting with antibodies against RhopH3 or RhopH1/Clag3.1 revealed the absence of RhopH3 from the immunoprecipitates derived from the rapamycin-treated parasites, although RhopH2 and RhopH1/Clag3.1 still showed association. Arrowheads indicate the expected position of migration of the full-length (WT) and truncated RhopH3, and RhopH1/Clag3.1. The *rhopH3-loxP* clone 5F5 was used throughout for these experiments.

The following figure supplement is available for figure 2:

**Figure supplement 1.** Truncation of RhopH3 leads to mistrafficking of components of the RhopH complex.

To better define the fate of the mistrafficked RhopH3△4–6 in the mutant parasites, rapamycin-treated mature schizonts were probed with antibodies to the merozoite plasma membrane surface marker MSP1. This indicated that the mutant protein was expressed in a location surrounding (and so likely external to) the plasma membrane of individual segmented intracellular merozoites (*Figure 2—figure supplement 1*). In confirmation of this, IFA of naturally released free merozoites showed that the truncated RhopH3△4–6 was often largely undetectable in merozoites of the mutant parasites (*Figure 2C*).

One interpretation of these results was that truncation of RhopH3 interfered with formation of the RhopH complex. To test this notion, we used a monoclonal antibody (mAb) specific for RhopH2 to immunoprecipitate the complex from extracts of schizonts of *rhopH3-loxP* clone 5F5. As shown in *Figure 2D*, both RhopH3 and RhopH1/Clag3.1 were precipitated as expected from lysates of control parasites. In contrast, RhopH3△4–6 was undetectable in the precipitate from lysates of rapamycin-treated parasites, although RhopH1/Clag3.1 could still be detected. This showed that truncation of RhopH3 ablates its association with RhopH2, although it does not appear to affect the interaction between RhopH2 and RhopH1/Clag3.1. Collectively, these results suggested that truncation of RhopH3 caused mistrafficking of other components of the complex, probably due to loss of the association between RhopH3 and these other proteins.

## Loss of the RhopH complex is a lethal event

The above results showed that whilst truncation of RhopH3 affected trafficking of the RhopH complex, it did not prevent schizont development in the erythrocytic growth cycle in which the parasites were treated with rapamycin (henceforth referred to as cycle 1). To evaluate the effects of RhopH3 modification and mistrafficking on longer-term parasite viability, we first exploited a recently developed assay in which parasite replication is assessed in 96-well microplates over a period of 5 – 7 erythrocytic cycles by visualization of the localized lysis of host erythrocytes in static cultures in 96-well microplates. Under these conditions, successful parasite growth results in formation of microscopically discernible zones of clearance of erythrocytes referred to as plaques (*Thomas et al., 2016*). As shown in *Table 1*, in three separate assays DMSO-treated *rhopH3-loxP* parasites seeded at ~10 parasites per well produced plaques in nearly every well, with a mean average of ~8 plaques per well for clone 5F5 and ~5 plaques per well for clone 4B11 (*Table 1*). In contrast, in the plates seeded with an identical density of rapamycin-treated parasites, only ~10% of the wells contained plaques and no well contained more than one plaque (*Figure 3A*, *Table 1*). Analysis by diagnostic PCR of several parasite clones isolated from individual plaques that appeared in plates seeded with rapamycin-treated parasites revealed that in all cases they derived from parasites that possessed an intact *rhopH3-loxP* gene, indicating that these corresponded to a small subpopulation of parasites in which excision of the floxed sequence had not taken place (*Figure 3B*). Further analysis by PCR of one of these non-excised clones showed that the DiCre cassette had been lost (*Figure 3C*) probably due to a genomic rearrangement. This parasite clone (named RhopH3 NE) served as a useful control for subsequent experiments.

To further examine the effects of RhopH3 truncation on long-term parasite viability, low parasitaemia cultures of the *rhopH3-loxP* clones 5F5 and 4B11 were divided equally into two flasks,

**Table 1.** Conditional truncation of RhopH3 results in decreased parasite survival as determined by plaque assay.

| *Plaque assay no. | Treatment | †Proportion of wells containing plaques (%) | Mean number of plaques/well |
|---|---|---|---|
| 1 (clone 5F5) | DMSO | 98.88 | 7.7 |
| | Rapamycin | 10.56 | 0.11 |
| 2 (clone 5F5) | DMSO | 100 | 9.1 |
| | Rapamycin | 8.89 | 0.09 |
| 3 (clone 4B11) | DMSO | 99.44 | 5.24 |
| | Rapamycin | 9.44 | 0.1 |

*Three independent plaque assays were set up on different days.
†A total of 180 wells were used for each treatment in each assay.

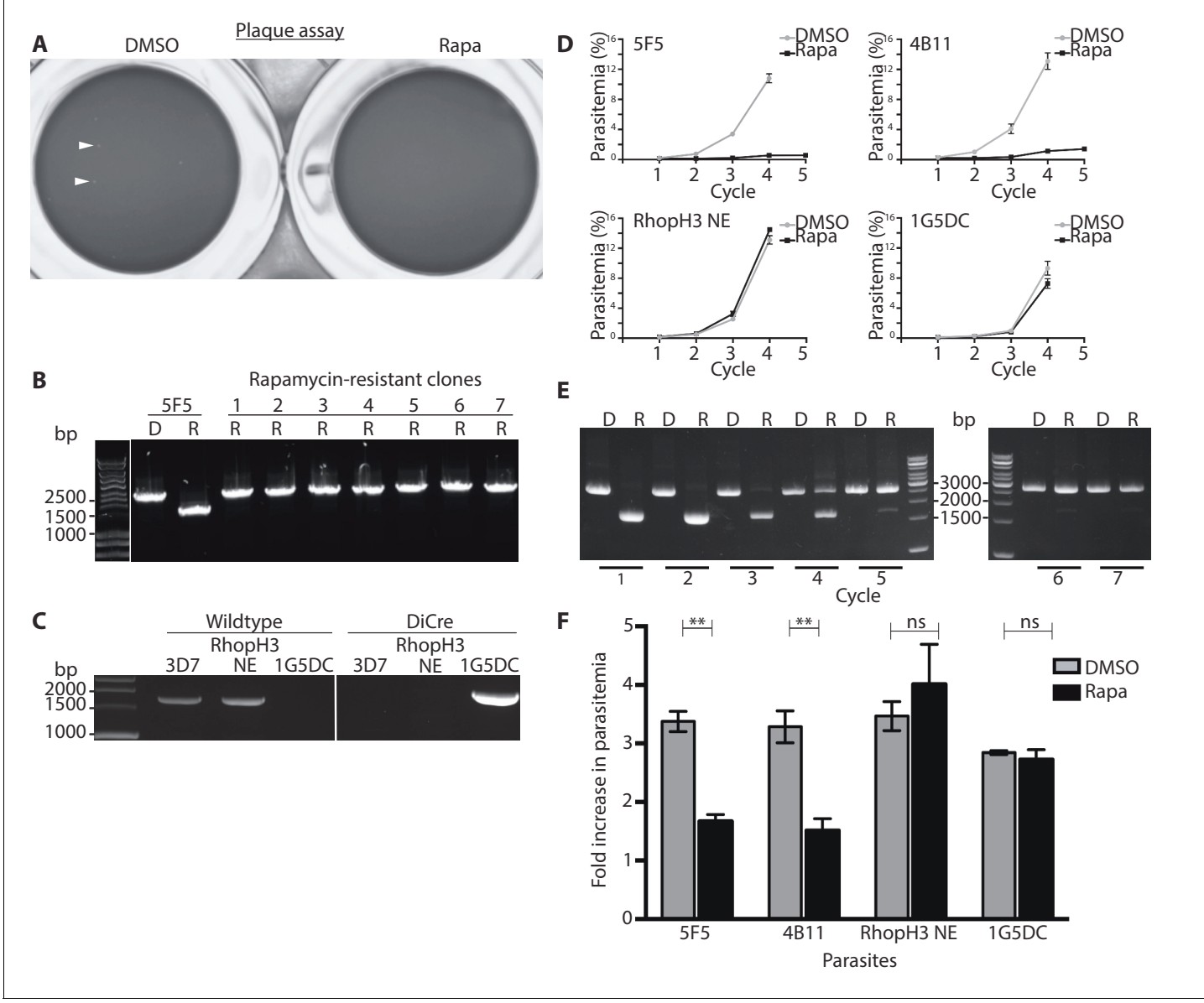

**Figure 3.** Loss of long-term viability in parasites lacking the RhopH complex. (A) Representative wells seeded with identical concentrations (10 parasitised cells/well) of DMSO-treated or rapamycin-treated *rhopH3-loxP* clone 5F5 parasites, showing formation of plaques only in the wells seeded with DMSO-treated parasites. Two of the plaques are indicated by white arrowheads. (B) PCR analysis of the *rhopH3-loxP* locus in the small number of clones isolated from wells seeded with rapamycin-treated *rhopH3-loxP* parasites. The size of the PCR product indicates excision of the floxed sequence had not taken place in these seven clones (numbered 1–7), whereas rapamycin induced efficient excision in the parent 5F5 clone (left-hand two tracks). For PCR strategy, see *Figure 1A*. (C) PCR analysis of the modified *SERA5* locus in parasite clone RhopH3 NE, showing loss of the DiCre cassette in this clone. (D) Growth curves showing replication of parasites of the indicated clones over the course of 5 erythrocytic cycles. Data were averaged from three biological replicate experiments and presented as the mean ± standard error of the mean (SEM). (E) Non-excised parasites quickly outgrow RhopH3△4–6 parasites. The relative abundance of parasites harbouring the excised or intact *rhopH3-loxP* locus in a population of rapamycin-treated *rhopH3-loxP* clone 5F5 parasites was determined by diagnostic PCR over the course of 7 erythrocytic growth cycles (indicated, where cycle 1 indicates that in which treatment occurred). (F) Decreased erythrocyte invasion by rapamycin-treated *rhopH3-loxP* parasites. Parasites of the indicated clones were treated with DMSO or rapamycin and allowed to invade fresh erythrocytes. Ring-stage parasitemia levels were determined 4 hr later. Data were averaged from three biological replicate experiments. Error bars depict standard error of the mean. Statistical significance was determined by a two-tailed t-test where p≤0.01 (indicated by asterisks) and p>0.05, non-significant (ns).

treated with either DMSO or rapamycin, then the parasites simply maintained in continuous culture, monitoring increase in parasitaemia at 48 hr intervals as well taking samples for analysis by diagnostic PCR. Cultures of the parental 1G5DC parasites as well as the DiCre-defective RhopH3 NE clone were similarly treated and monitored in parallel. As shown in *Figure 3D*, whilst replication of the 1G5DC and RhopH3 NE parasites was unaffected by rapamycin treatment, the rapamycin-treated 5F5 and 4B11 clones showed a dramatic decrease in growth rate. However in both clones the appearance of replicating parasites was evident by cycle 3, suggesting that these might correspond to a minor population of normally-replicating non-excised parasites. Diagnostic PCR analysis of the 5F5 culture supported this notion. At the end of cycle 1, PCR using primers that distinguish between the excised and non-excised locus showed the expected highly efficient excision of the floxed *rhopH3-loxP* sequence in the rapamycin-treated culture, with the non-excised locus undetectable. However, periodic examination of the parasites by diagnostic PCR over the ensuing 6 erythrocytic cycles showed a time-dependent increase in the proportion of non-excised parasites in the rapamycin-treated culture, suggesting that the initially undetectable population of non-excised parasites gradually overgrew the cultures. This occurred likely as a result of a selective advantage conferred on them by the replication defect displayed by the RhopH3△4–6 parasites. By cycle 5, the excised locus was hardly detectable in the rapamycin-treated culture (*Figure 3E*). Together with the results of the plaque assay, these results allowed us to conclude that truncation of RhopH3 results in complete loss of long-term parasite viability.

## Loss of the RhopH complex leads to an invasion defect

The severe growth defect displayed in the plaque and growth assays could result from an inability of mutant parasites to egress from the host erythrocyte, a block in invasion, or a developmental arrest during intracellular growth. We therefore next investigated the capacity of RhopH3△4–6 parasites to undergo egress. For this, we used time-lapse differential interference contrast (DIC) microscopy to observe the egress of merozoites from highly mature, synchronized schizonts at the end of cycle 1 (that is, ~45 hr following treatment of ring-stage *rhopH3-loxP* parasites with rapamycin or DMSO). This revealed no gross differences in the efficiency or morphology of egress (*Video 1*), indicating that the absence of the RhopH complex from rhoptries does not affect egress.

To investigate a potential invasion phenotype resulting from RhopH3 truncation, a synchronized culture of *rhopH3-loxP* parasites at early ring stage was divided into two, treated with either DMSO or rapamycin and then allowed to mature to schizont stage before purifying the mature schizonts and adding them to fresh erythrocytes. After incubation for a further 4 hr to allow the *rhopH3-loxP* schizonts to undergo merozoite egress and invasion, the percentage of erythrocytes infected with cycle 2 ring-stage parasites was quantified. The results consistently showed that the ring-stage parasitemia values in cultures derived from the rapamycin-treated *rhopH3-loxP* parasites was only ~50% of that in their DMSO-treated counterparts (*Figure 3F*). Importantly, invasion by the control RhopH3 NE and the parental 1G5DC parasites was unaffected by rapamycin treatment. Taken together with the other results, these data showed that the absence of the RhopH complex from parasite rhoptries significantly affects the ability of the parasite to invade new host cells.

## The RhopH complex is required for intracellular parasite development

Although the results of the above experiments pointed to a severe invasion defect in parasites lacking the RhopH complex, it was unclear

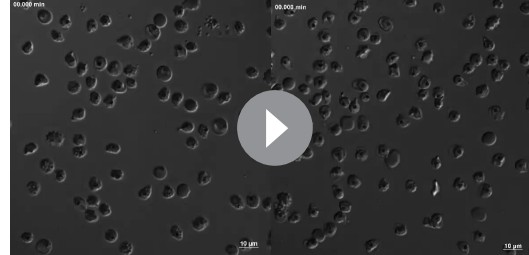

**Video 1.** Parasite egress is unaffected by loss of the RhopH complex. Synchronized parasites of *rhopH3-loxP* clone 4B11 were treated with DMSO or rapamycin at ring stage, then allowed to mature to schizont stage and further synchronised by incubation for 3–5 hr in the presence of 1 μM (4-[7-[(dimethylamino)methyl]−2-(4-fluorphenyl)imidazo[1,2-α]pyridine-3-yl]pyrimidin-2-amine (compound 2), which reversibly stalls egress. Egress of the parasites was then monitored by time-lapse DIC video microscopy following removal of the compound 2, as described previously (*Das et al., 2015*). DMSO-treated samples are shown on the left, rapamycin-treated are samples shown on the right.

whether this could be sufficient to explain the results of the plaque assay, which indicated a complete lack of long-term viability in the RhopH3△4–6 mutants. To explore this further, we examined growth and development of the mutants using microscopic examination of Giemsa-stained cultures. This showed that whereas rapamycin-treated *rhopH3-loxP* parasites appeared morphologically normal at the end of the cycle 1 as well as at the ring stage of cycle 2, development of the mutant parasites stalled at trophozoite stage in cycle 2 (*Figure 4A*) and the parasites did not develop into schizonts. To confirm this developmental block we used flow cytometry to monitor the DNA content of the parasites in cycle 2. This confirmed that rings derived from rapamycin-treated *rhopH3-loxP* parasites did not increase their DNA content during cycle 2 (*Figure 4B*), that is, they did not progress to the multinuclear schizont stage. Taken together, these data indicated that the RhopH complex is essential for the trophozoite to schizont developmental transition of the intracellular parasite.

## Protein export occurs normally in the RhopH3△4–6 mutants

Export of parasite proteins into the host erythrocyte is important for parasite virulence and for the uptake of nutrients; blocking export prevents modification of the erythrocyte surface with the knob structures that play a role in cytoadhesion, and also prevents development of the parasite beyond the trophozoite stage (*Beck et al., 2014*; *Elsworth et al., 2014*). Since we observed a similar growth phenotype in cycle 2 in the RhopH3△4–6 parasites, we decided to determine whether the developmental arrest was the result of a generalized defect in protein export. To do this, we examined the subcellular localization of KAHRP and MAHRP1, parasite proteins that are established markers for protein export and Maurer's clefts (intraerythrocytic membranous structures of parasite origin) respectively, in cycle 2 trophozoites derived from rapamycin-treated *rhopH3-loxP* parasites (*Crabb et al., 1997*; *Spycher et al., 2008*). This revealed no discernible alterations in protein export and Maurer's cleft formation in the RhopH3△4–6 mutants (*Figure 5A*). This conclusion was corroborated by electron microscopy, which revealed the formation of knobs on the surface of erythrocytes infected with rapamycin-treated *rhopH3-loxP* parasites (*Figure 5B*). We concluded that protein export from the intracellular parasite can take place normally in the absence of the RhopH complex.

## Import pathways are defective in *rhopH3* mutant parasites

The developmental arrest observed in cycle 2 trophozoites of the RhopH3△4–6 parasites was strikingly reminiscent of the effect of isoleucine starvation on *P. falciparum* (*Babbitt et al., 2012*). Isoleucine is transported into the parasitized cell via the PSAC/NPP, the parasite-induced uptake pathway responsible for enhanced nutrient uptake in parasite-infected erythrocytes (*Martin and Kirk, 2007*). The PSAC/NPP is also responsible for the permeability of parasite-infected erythrocytes to the alcohol sugar sorbitol (*Nguitragool et al., 2011*), leading to the capacity of sorbitol solutions to mediate osmotic lysis of infected erythrocytes. This lysis can be readily quantified by measuring levels of host cell hemoglobin released following treatment of parasitized cells with a sorbitol solution (*Pillai et al., 2010*). To determine whether the PSAC/NPP was functional in the RhopH3△4–6 mutants, their resistance to sorbitol-mediated lysis was assessed. As shown in *Figure 6A*, erythrocytes infected with parental 1G5DC parasites or the non-excised RhopH3 NE clone displayed the expected sensitivity to sorbitol, as did erythrocytes infected with control (DMSO-treated) *rhopH3-loxP* parasites. In contrast, erythrocytes infected with cycle 2 rapamycin-treated *rhopH3-loxP* parasites were insensitive to sorbitol; the amount of hemoglobin released upon sorbitol treatment was not significantly different from the amount released by treatment of the infected erythrocytes with an isotonic control buffer (PBS).

To further investigate the activity of the PSAC/NPP in the RhopH3△4–6 mutants, erythrocytes infected with cycle 2 rings were incubated with 5-aminolevulinic acid (5-ALA). This compound is excluded from uninfected erythrocytes but is taken up by infected erythrocytes and converted to fluorescent protoporphyrin IX (PPIX) (*Sigala et al., 2015*). Import of 5-ALA has previously been shown to be sensitive to furosemide, a small molecule inhibitor of PSAC/NPP, and is also blocked upon downregulation of parasite export and PSAC/NPP activity in transgenic *P. falciparum* (*Beck et al., 2014*; *Sigala et al., 2015*; *Staines et al., 2004*). Import of 5-ALA therefore acts as a convenient reporter for PSAC/NPP activity. Examination by fluorescence microscopy (*Figure 6B and C*) and flow cytometry (*Figure 6D*) showed that erythrocytes infected with DMSO-treated *rhopH3-loxP* clone 5F5 and 4B11 parasites readily took up 5-ALA, whereas no fluorescence was observed in erythrocytes infected with rapamycin-treated *rhopH3-loxP* parasites following incubation with 5-

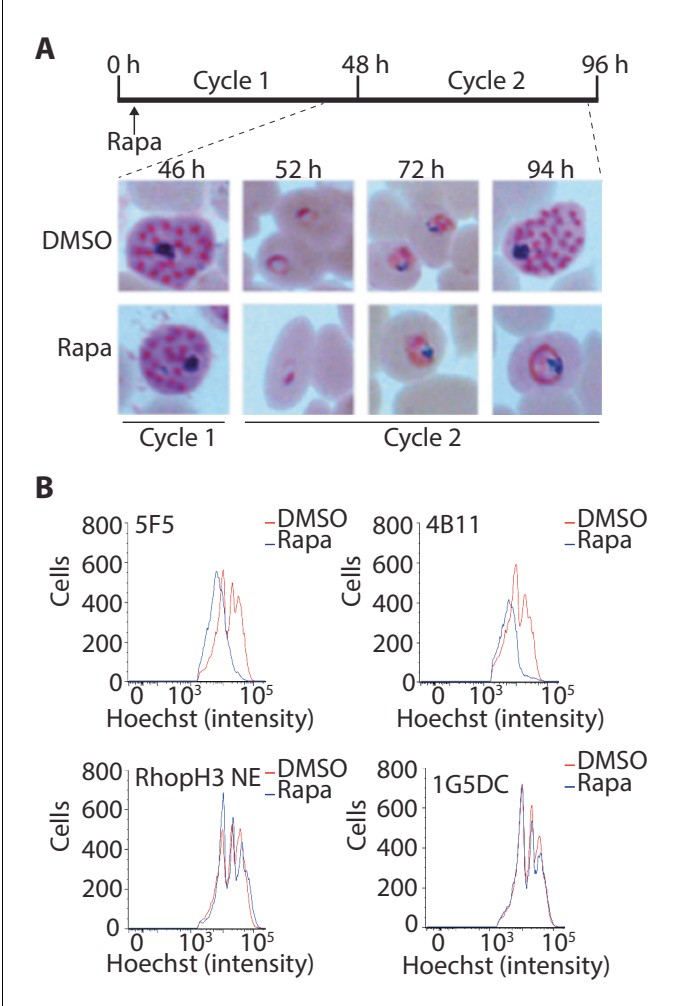

**Figure 4.** Loss of the RhopH complex results in developmental arrest. (**A**) Developmental block in rapamycin-treated *rhopH3-loxP* parasites. Giemsa-stained images showing intracellular development of DMSO-treated and rapamycin-treated *rhopH3-loxP* clone 5F5 parasites from the end of cycle 1 to the end of cycle 2. A clear developmental block was evident in the rapamycin-treated parasites in cycle 2. The number of hours following the beginning of cycle 1 is indicated, as well as its relation to the time point of rapamycin treatment (indicated in the schematic timeline). (**B**) Flow cytometry analysis of DMSO-treated and rapamycin-treated *rhopH3-loxP* clones 5F5 and 4B11. Analysis was performed at the end of cycle 2 (92 hr after rapamycin-treatment). The intensity of Hoechst 33342 staining provides a measure of the DNA content of the parasites, reflecting parasite development.

ALA. In contrast, rapamycin-treatment had no effect on the capacity of the parental 1G5DC or DiCre-deficient RhopH3 NE parasites to take up 5-ALA (*Figure 6B–D*). Combined, these results convincingly indicate that the PSAC/NPP is defective in the *rhopH3* mutants.

## Discussion

In this study we have shown that RhopH3 plays a central role in the formation of a functional RhopH complex and that mutation of RhopH3 results in two severe, but very distinct, phenotypes: (1) a ~50% decrease in host erythrocyte invasion; and (2) a block in the development in the early trophozoite stage of those parasites that do invade. This block in development is a lethal event; no parasites carrying the mutant form of the *rhopH3-loxP* gene were recovered in a plaque assay and parasites with an intact *rhopH3-loxP* gene quickly outgrew the mutant parasites after rapamycin treatment. These results represent the first published evidence that RhopH3 is essential and mark

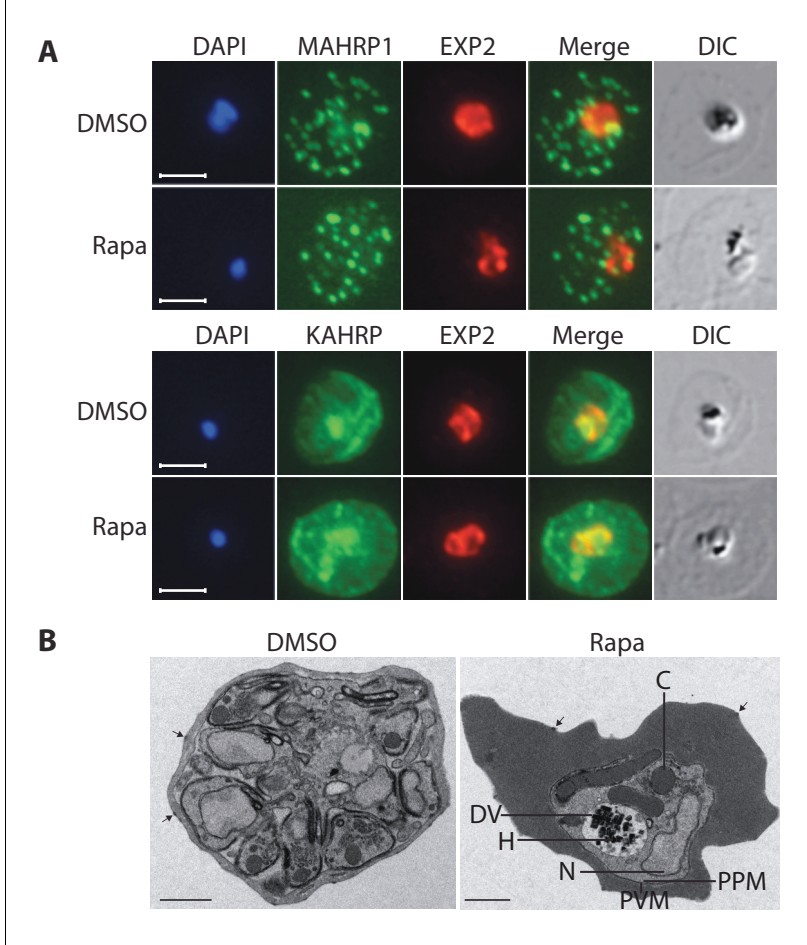

**Figure 5.** Loss of the RhopH complex does not ablate parasite protein export. Cycle 2 (72 hr post rapamycin treatment) DMSO-treated and rapamycin-treated *rhopH3-loxP* clone 5F5 trophozoite-stage parasites were probed with antibodies against the parasitophorous vacuole membrane marker EXP2 to delineate the parasite in the infected erythrocyte, as well as antibodies specific for either the Maurer's cleft marker MAHRP1 (top panels) or the export marker KAHRP (bottom panels). Scale bar, 5 µm. (**B**) Transmission electron micrograph showing a comparison between cycle 2 parasites of DMSO-treated or rapamycin-treated *rhopH3-loxP* clone 5F5 parasites ~92 hr following rapamycin treatment. The developmental block in the RhopH3△4–6 parasite is clearly evident, as is the presence of knobs (arrowed) on the surface of the erythrocyte in both cases. Components of the mutant parasite labelled are the digestive vacuole (DV), haemozoin (H), nucleus (N), parasitophorous vacuole membrane (PVM), cytostomes (**C**) and parasite plasma membrane (PPM). The mutant parasites displayed no obvious ultrastructural differences from wild type trophozoites at a similar developmental stage (not shown). Scale bar, 1 µm.

the first time a rhoptry protein has been shown to have two separate, seemingly unrelated functions at different stages of the erythrocytic life cycle. RhopH3 is also the first rhoptry bulb protein to be directly assigned a role in invasion; other rhoptry proteins previously experimentally implicated in invasion are located in the rhoptry neck. The release of rhoptry neck proteins is considered the step at which the parasite commits to host cell entry (*Srinivasan et al., 2011*), so the discovery of an important invasion factor that is presumably released later in the invasion pathway is important.

Whilst it was surprising that loss of the function of the RhopH complex leads to two different, seemingly unrelated, phenotypes, previous results had hinted at a role for RhopH3 and the complex in both processes. RhopH3, and proteolytic fragments of RhopH3, can bind to erythrocytes and to liposomes (*Sam-Yellowe and Perkins, 1990*, *1991*). This appears to occur even in the absence of other members of the RhopH complex, indicating that, for its role in invasion, RhopH3 may not

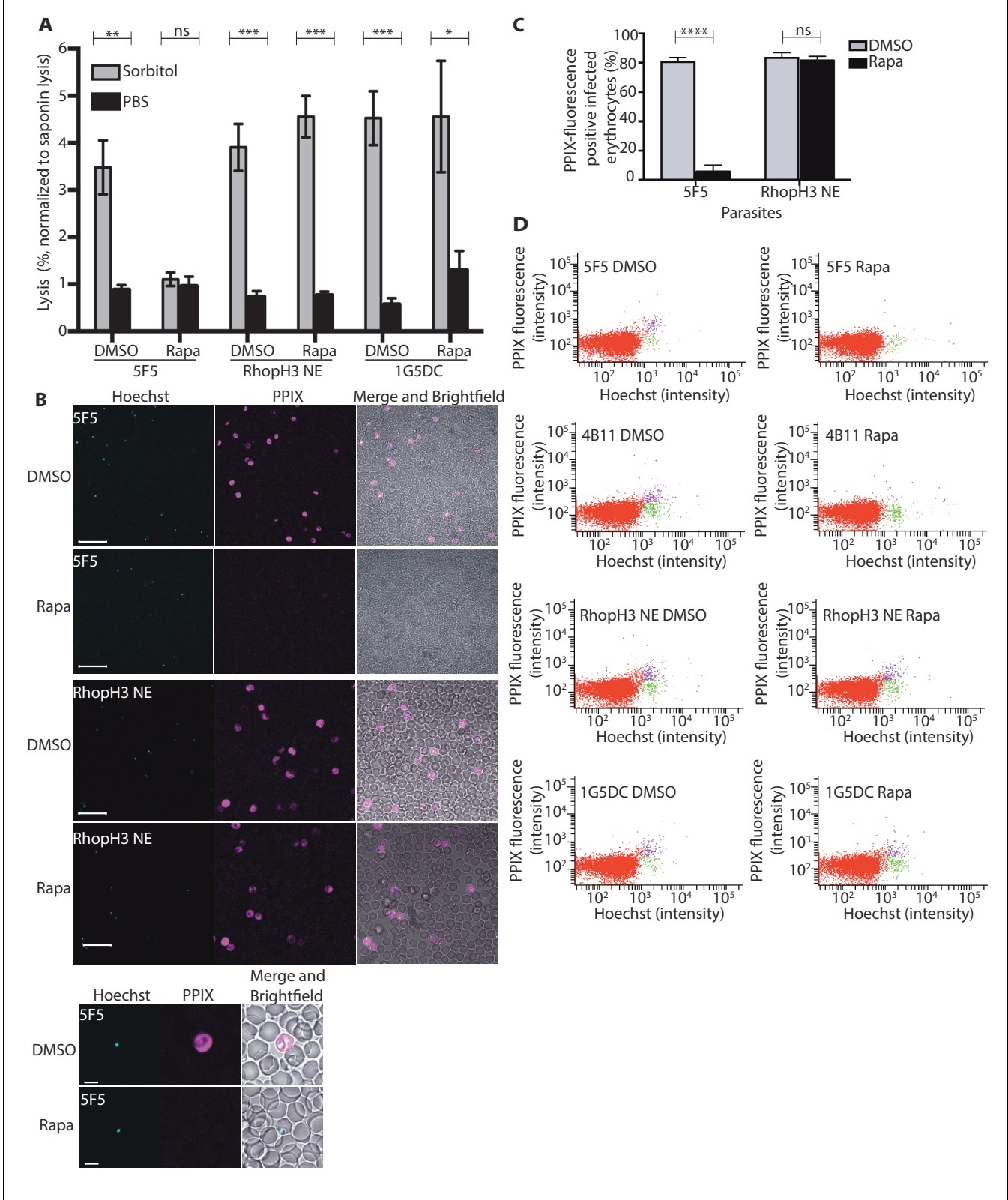

**Figure 6.** Loss of the RhopH complex results in reduced sorbitol sensitivity and reduced uptake of exogenous small molecules. (A) Synchronous cycle 2 parasites of the indicated clone (parasitaemia ~5%) treated 72 hr previously with DMSO or rapamycin in cycle 1 were suspended in osmotic lysis buffer containing 280 mM sorbitol or in PBS, and the resulting cell lysis determined by measuring the absorbance of the supernatant at 405 nm. An equal volume of parasite culture was lysed in 0.15% (w/v) saponin to give a value for 100% lysis and all other absorbance values normalized to this. Data were

*Figure 6 continued on next page*

Figure 6 continued

averaged from three biological replicate experiments. Statistical significance was determined by a two-tailed t-test; significance levels are indicated: p≤0.001, ***; p≤0.01, **;p≤0.05, *; and p>0.05, non-significant (ns). (B) Uptake of 5-ALA by erythrocytes infected with either DMSO-treated or rapamycin-treated *rhopH3-loxP* clone 5F5 parasites at cycle 2. Cultures were incubated overnight with 200 µM 5-ALA and uptake of the compound and its subsequent conversation to PPIX in infected erythrocytes visualized by fluorescence microscopy. Infected erythrocytes were visualized by staining with Hoechst 33342. Top panels show fields of view containing multiple infected erythrocytes of the indicated strain. Scale bar, 50 µm. Bottom panels show individual infected erythrocytes. Scale bar, 5 µm. (C) Quantitation of the levels of uptake of 5-ALA by infected erythrocytes. For each condition, a total of 1300 Hoechst-positive cells were analyzed for intensity of PPIX fluorescence using MetaMorph (Molecular Devices) and a statistical significance was determined by a two-tailed t-test. Significance levels are indicated: p≤0.0001, **** and p>0.05, non-significant (ns). (D) Flow cytometry analysis of 5-ALA-treated parasites. Uptake of 5-ALA and its subsequent conversation to PPIX in cycle 2 parasites following treatment in cycle 1 with rapamycin or DMSO was determined by flow cytometry of Hoechst stained parasites. Gating was applied to distinguish Hoechst negative cells (red population), Hoechst positive/PPIX negative cells (green population) and Hoechst positive/PPIX-positive cells (purple population). For the 1G5DC parental and RhopH3-loxP NE parasite clones, most of the parasites were positive for both Hoechst and PPIX fluorescence regardless of their treatment with rapamycin or DMSO. In contrast, for rapamycin-treated *rhopH3-loxP* clones 5F5 and 4B11, most of the parasites were Hoechst positive/PPIX negative indicating a defect in 5-ALA uptake.

require the function of the other proteins of the complex. Further supporting a role for RhopH3 in invasion is the finding that anti-RhopH3 antibodies can block invasion (*Doury et al., 1994*; *Sam-Yellowe and Perkins, 1991*). Nonetheless it is curious that ~50% of the parasites still enter the erythrocyte in the absence of full-length RhopH3. RhopH3△4–6 may retain sufficient activity in 50% of the parasites to allow invasion to take place. Alternatively, these parasites could use a RhopH3-independent pathway. Invasion by *P. falciparum* relies on several redundant pathways and there is precedent for a partial reduction of invasion by mutant parasites (*Tham et al., 2012*). It will be of interest to determine whether invasion pathways that are currently thought to be redundant become essential in the absence of wild type RhopH3. A third possibility is that RhopH3 is involved in a post-invasion process. However, the invasion assay used here would detect all intracellular parasites that have completed invasion. We therefore favor the interpretation that the observed decrease in the number of newly invaded *rhopH3-loxP* parasites indicates that the invasion process is not completed in the mutants.

The other phenotype displayed by the RhopH3△4–6 mutants, the block in development during the early trophozoite stage, is likely the result of a defect in nutrient intake owing to improper trafficking ablating the function of RhopH1/Clag. The loss of sorbitol sensitivity of erythrocytes infected with the RhopH3△4–6 mutants and their impermeability to 5-ALA indicate that RhopH1/Clag3.1 (the RhopH1/Clag3 isotype expressed in these 3D7-derived parasites) and RhopH1/Clag3.2 are not functioning at the erythrocyte surface. RhopH1/Clag3 proteins are transported to the erythrocyte surface in a PEXEL-independent manner (*Beck et al., 2014*) and are exposed on the surface of the erythrocyte (*Nguitragool et al., 2014*), but the mechanism by which these proteins are transported from the rhoptry, beyond the PVM and to the erythrocyte plasma membrane is unknown. RhopH3 and RhopH2 are have been detected in the PVM immediately after invasion, (*Sam-Yellowe et al., 1988*; *Hiller et al., 2003*), as well as in the erythrocyte at later developmental stages of the intraerythrocytic parasite (*Beck et al., 2014*; *Vincensini et al., 2008*). It is likely that mislocalization of RhopH3△4–6 in merozoites prevents the proper, or properly timed, release of the complex during invasion and prevents RhopH1/Clag from being delivered to its correct location. As the RhopH1/Clag3.1 and RhopH1/Clag3.2 in the trophozoite stage parasite derives entirely from protein that is introduced during invasion (*Ling et al., 2003*), mislocalization at the merozoite stage likely cannot be corrected by additional subsequent protein synthesis.

The complete arrest in development of the RhopH3△4–6 mutants in the cycle following gene modification (cycle 2) also provides insight into the potential roles of the RhopH1/Clag proteins. Little is known about these proteins other than the importance of the RhopH1/Clag3.1 and RhopH1/Clag3.2 proteins in the PSAC (*Nguitragool et al., 2011*). However, chemical inhibition of RhopH1/Clag3.2 function and PSAC activity in parasites cultured in rich medium (RPMI 1640, the same medium used in this study) leads to only a small decrease in parasite viability (*Pillai et al., 2012*). Similarly, parasites that do not produce RhopH1/Clag3.1 or RhopH1/Clag3.2 have only a minimal growth disadvantage compared to wild type parasites in a competition assay (*Comeaux et al., 2011*). Parasites lacking RhopH1/Clag9 have no apparent growth phenotype *in vitro* (the gene is

absent from the D10 and T9-96 laboratory strains that lack part of chromosome 9) (*Chaiyaroj et al., 1994*; *Day et al., 1993*). In contrast, parasites lacking RhopH1/Clag2 or RhopH1/Clag8 have not been reported so the essentiality of these proteins is unknown. We speculate that the RhopH3△4–6 mutants very likely transport none of the RhopH1/Clag proteins to the erythrocyte surface. If so, the observed growth phenotype is therefore essentially that of a disruption of all the *clag* genes. The striking growth phenotype of this mutant is in stark contrast to the mild phenotype of RhopH1/Clag3.2 inhibition (*Pillai et al., 2012*) or the absence of both the RhopH1/Clag3 proteins (*Comeaux et al., 2011*) when the parasites are grown in RPMI. This may indicate that RhopH1/Clag2 and RhopH1/Clag8 play important roles in nutrient uptake as well, as previously suggested (*Pillai et al., 2012*), and that RhopH complexes containing several different RhopH1/Clag proteins together mediate the uptake of the nutrients required for parasite growth in the infected erythrocyte. Interestingly, most *Plasmodium* species encode fewer RhopH1/Clag proteins than *P. falciparum*; some species possess only two *clag* genes, comprising a *clag9* orthologue and a second orthologue more closely related to the other *P. falciparum clag* genes (*Counihan et al., 2013*; *Kaneko et al., 2001*). In conclusion, our results raise the intriguing possibility that, in *P. falciparum*, other RhopH1/Clag proteins, in addition to RhopH1/Clag3.1 and RhopH1/Clag3.2, function in nutrient import.

Together the results presented in this study provide new insights into the role of the rhoptry in the malarial blood stages and reveal that rhoptry proteins can function in multiple, distinct processes. They furthermore show that the functions of rhoptry proteins extend beyond the initial invasion of the erythrocyte and can affect parasite growth throughout the blood stage life cycle.

## Note added in proof

A manuscript describing an alternative strategy to deplete *Plasmodium falciparum* of RhopH3 was recently published by Ito and colleagues (*Ito et al., 2017*). These authors derive very similar conclusions about the role of RhopH3 in the invasion process and nutrient uptake.

## Materials and methods

### Reagents and antibodies

Oligonucleotide primers were from Sigma-Aldrich (Gillingham, United Kingdom), as was rapamycin (cat# R0395), which was prepared as a 10 µM stock in DMSO. 5-aminoleuvlinic acid (5-ALA) from Sigma-Aldrich (cat# A3785) was prepared as a 1 mM stock in DMSO. Restriction enzymes were from New England Biolabs (Ipswich, MA). The antifolate drug WR99120 (Jacobus Pharmaceuticals, Princeton, NJ), was stored as a 20 µM stock in DMSO. Polyclonal antiserum α-Ag44, which recognizes the C-terminal 134 amino acid residues of RhopH3 (*Coppel et al., 1987*), was a kind gift of Ross Coppel (Monash University, Australia). A polyclonal antiserum against *P. falciparum* AMA1 has been previously described (*Collins et al., 2009*), as have polyclonal antibodies against *P. falciparum* MSP1 and the anti-MSP1 mAb 89.1 (*Holder et al., 1985b*). Other antibodies were kindly provided by Osamu Kaneko, Nagasaki University Japan (rabbit anti-RhopH1/Clag3.1), John Vakonakis, University of Oxford UK (rabbit anti-MAHRP1), Ross Coppel at Monash University Australia (anti-KAHRP), Tony Holder, the Francis Crick Institute UK (anti-RhopH2 mAb 61.3). Monoclonal antibody 7.7 (anti-EXP2) was from The European Malaria Reagent Repository, contributed by Jana McBride, and the mouse anti-RAP2 mAb MRA-876 was obtained from BEI resources, National Institute of Allergy and Infectious Diseases (NIAID), National Institutes of Health (NIH), contributed by Allan Saul. Use of these antibodies in immunoblot and IFA analyses have been described elsewhere (*Holder et al., 1985a*; *Kaneko et al., 2005*; *Coppel et al., 1987*; *Collins et al., 2009*; *Hall et al., 1983*; *Oberli et al., 2014*; *Pei et al., 2005*; *Saul et al., 1992*).

### *P. falciparum* culture, transfection and growth analysis

All *P. falciparum* transgenesis work described used the 3D7-derived DiCre-expressing clone 1G5DiCre (*Collins et al., 2013*), here referred to as 1G5DC. Asexual blood stage parasites were continuously cultured in RPMI 1640 medium containing Albumax (Gibco, Grand Island, NY) as a serum substitute and synchronised using established procedures (*Blackman, 1994*). For introduction of transfection plasmids, mature schizonts were enriched using Percoll (GE Healthcare) and

electroporated using an Amaxa 4D electroporator and P3 Primary cell 4D Nucleofector X Kit L (Lonza, Basel, Switzerland) using programme FP158 as described (*Collins et al., 2013*).

Long-term parasite growth as measured by plaque-forming ability was determined by diluting trophozoite-stage cultures to a density of 10 parasites per well in complete medium with human erythrocytes at a haematocrit of 0.75% and plating 200 μL of this suspension into flat bottomed 96 well microplates, as previously described (*Thomas et al., 2016*). Plates were incubated for 10 days in gassed humidified sealed modular chambers before plaque formation was assessed by microscopic examination using a Nikon TMS inverted microscope (40x magnification) and documented using a Perfection V750 Pro scanner (Epson, Long Beach, CA).

Growth characteristics of mutant parasites were determined by microscopy of Giemsa-stained thin films. Long-term growth was also measured using flow cytometry of hydroethidine-stained trophozoite-stage parasites, as described (*Stallmach et al., 2015*). Cultures adjusted to a parasitaemia of 0.1% were monitored every 48 hr for up to seven intraerythrocytic cycles. All experiments were carried out in triplicate, data analysed using GraphPad Prism and presented as the mean ± standard error of the mean (SEM). In addition, cells 92 hr post-rapamycin treatment were fixed in 4% formaldehyde and 0.1% glutaraldehyde and stained with 2 μM Hoechst 33342 (Invitrogen, Waltham, MA) before detection of the Hoechst emission (a measure of DNA replication) by the 355 nm laser of a LSR II (BD Biosciences, San Jose, CA), through a 440/40 nm bandpass filter. Doublet cells were excluded using a FCS-A versus FCS-H display. Samples were analysed using FlowJo software.

Erythrocyte invasion assays were carried out using a modification of a method described previously (*Moss et al., 2012*). Highly synchronous, mature schizonts of the parasite clones under examination were enriched by centrifugation over Percoll cushions then added at a parasitaemia of 1% to fresh erythrocytes. After incubation for 4 hr, parasites were stained with SYBR Green-I and the percentage of newly ring-infected erythrocytes was determined by flow cytometry using a BD FACS Calibur flow cytometer (BD Biosciences). SYBR Green-I was excited by a 488 nm 20 mW blue laser and detected by a 530/30 filter. BD CellQuest Pro (BD Biosciences, UK) was used to collect 100,000 events per sample. Experiments were done in triplicate, data analysed using GraphPad Prism and presented as the mean ± SEM.

## Generation of *rhopH3-loxP* parasites and conditional RhopH3 truncation

Parasites harbouring a floxed segment of the genomic *rhopH3* gene were generated by Cas9-mediated replacement of *rhopH3* endogenous introns 3 and 6 as well as the intervening sequence. The repair plasmid, called pESS_R3_E46_loxP (synthesised by GENEWIZ, South Plainfield, NJ) comprised synthetic heterologous *loxP*-containing *SERA2* and *sub2* introns (*Jones et al., 2016*) flanking a recodonized form of *rhopH3* exons 4–6. The complete native sequences of exons 3 and 7 were included on either side of this central module to act as flanking regions for homology-directed repair. Protospacer Workbench (*MacPherson and Scherf, 2015*) was used to identify 20 bp protospacer sequences specifically targeting *rhopH3*. To generate pSgRNA plasmids expressing suitable sgRNAs, InFusion-based cloning (Clontech, Mountain View, CA) was used to replace the BtgZI adaptor sequence of pL6-X (*Ghorbal et al., 2014*) with annealed oligos encoding a sgRNA targeting *rhopH3* exon 4 (RHOPH3_sgRNA_E4_F and RHOPH3_sgRNA_E4_R, generating pSgRNA1), 5 (RHOPH3_sgRNA_E5_F and RHOPH3_sgRNA_E5_R, generating pSgRNA2) or 6 (RHOPH3_sgRNA_E6_F and RHOPH3_sgRNA_E6_R, generating pSgRNA3) (see *Table 2* for sequences of all oligonucleotide primers used in this study). Schizonts of *P. falciparum* clone 1G5DC were transfected with 20 μg Cas9-expressing pUF1 (*Ghorbal et al., 2014*), 20 μg pESS_R3_E46_loxP repair plasmid and 8 μg of sgRNA-containing pSgRNA1, pSgRNA2 or pSgRNA3. Twenty-four hours post-transfection, the electroporated parasites were treated with 2.5 nM WR99210 for 48 hr to select for transfectants harbouring pUF1 before returning the cultures to medium without drug. Integrant parasites generally reached parasitaemia levels suitable for cryopreservation within 2–5 weeks.

Detection of integration of pESS_R3_E46_loxP in the parasite population was performed by diagnostic PCR using primers RHOPH3_exon2_F1 plus RHOPH3_exon5_WT_R (producing a product specific to the wild type *rhopH3* locus), or RHOPH3_exon2_F1 plus RHOPH3_exon4-6rec_R and RHOPH3_exon4-6rec_F plus RHOPH3_3UTR_R (producing products specific to the *rhopH3-loxP* modified locus). Integrant parasite clones *rhopH3-loxP* 5F5 and *rhopH3-loxP* 4B11 were then isolated by limiting dilution. Persistence of the integrated DiCre locus in these clones was confirmed by

**Table 2.** Oligonucleotide primers used in this study. Guide sequences shown in bold.

| Primer name | Sequence (5'−3') |
|---|---|
| RHOPH3_sgRNA_E4_F | taagtatataatatt**TTCTTCGTTTTTAAAAAAAG**gttttagagctagaa |
| RHOPH3_sgRNA_E4_R | ttctagctctaaaac**CTTTTTTTAAAAACGAAGAA**aatattatatactta |
| RHOPH3_sgRNA_E5_F | taagtatataatatt**CACCGATTTTAGCTTTAAAG**gttttagagctagaa |
| RHOPH3_sgRNA_E5_R | ttctagctctaaaac**CTTTAAAGCTAAAATCGGTG**aatattatatactta |
| RHOPH3_sgRNA_E6_F | taagtatataatatt**ACATTCTTATCATTATATTT**gttttagagctagaa |
| RHOPH3_sgRNA_E6_R | ttctagctctaaaac**ACATTCTTATCATTATATTT**aatattatatactta |
| RHOPH3_exon2_F1 | AGGAAATGGCCCAGACGC |
| RHOPH3_exon5_WT_R | TCTTTAAAGCTAAAATCGGTGATATTATGGCTC |
| RHOPH3_exon4-6rec_R | CAGGAAGTTACCTTTCAGCAGGG |
| RHOPH3_exon4-6rec_F | CCCTGCTGAAAGGTAACTTCCTG |
| RHOPH3_3UTR_R | CGAATATGTAATCAGTTGTATTTTTTCTCTAAAAGTTCATAG |
| +27 | CAATATCATTTGAATCAAACAGTGGT |
| −11 | CTTTGCCATCCAGGCTGTTC |
| −25 | CCATTGGACTAGAACCTTCAT |
| RHOPH3_exon7_R | CATAAAGAACGTCTTGTTTTCTGTATCCAATACC |
| RHOPH3_exon3_SB_F | CAAATATGCTATATGTGTAGGTACTCAATTTAAC |
| RHOPH3_exon3_SB_R | CATATAACTTTGGAGATGTAGAACCACAAGG |

PCR analysis using primers +27 plus −11 producing a 1900 bp product specific to the integrated DiCre cassette in 1G5DC parasites, or +27 plus −25 producing an amplicon of 1700 bp specific to the unmodified *SERA5* locus.

Recombination between the *loxP* sites was induced in tightly synchronised ring-stages of parasite clones *rhopH3-loxP* 5F5 and *rhopH3-loxP* 4B11 by incubation for 4 hr in the presence of 100 nM rapamycin in 1% (v/v) DMSO; mock treatment was with 1% (v/v) DMSO only (vehicle control). DiCre-mediated excision of the floxed *rhopH3* exons 4–6 was detected by PCR analysis of schizont stage genomic DNA (harvested ~40 hr following mock or rapamycin treatment) using primers RHOPH3_exon2_F1 and RHOPH3_exon7_R. Truncation of RhopH3 was evaluated by immunoblot analysis of SDS extracts of mature Percoll-enriched schizonts, probing with anti-Ag44 antibodies (or anti-AMA1 as a loading control) followed by horseradish-peroxidase secondary antibodies as described (*Jean et al., 2003*).

## Southern blot

For Southern blot analysis, a 738 bp probe corresponding to part of *rhopH3* exon 3 was produced by PCR amplification from *P. falciparum* IG5DC genomic DNA with primers RHOPH3_exon3_SB_F and RHOPH3_exon3_SB_R (*Table 2*). Probe radiolabelling and hybridisation to SacI/BsgI/XmnI-digested gDNA from clones of interest was as previously described (*Ruecker et al., 2012*).

## Immunoprecipitation and immunoblot analysis

For analysis of RhopH complex formation, mature Percoll-enriched schizont-stage parasites (42 hr post rapamycin or mock treatment) were harvested and stored at −80°C. Frozen parasite pellets were thawed into 100 µL NP-40 lysis buffer (1% Nonidet P-40, 150 mM NaCl, 50 mM Tris-HCL, 5 mM EDTA, 5 mM EGTA pH 8.0) containing a complete protease inhibitor cocktail (Roche, Indianapolis, IN). Samples were clarified by centrifugation at 14,000 rpm at 4°C and the supernatant passed through a 0.22 µm cellulose acetate Spin-X centrifuge tube filter (Corning Inc, Corning, NY). 100 µL Protein G-Sepharose beads (Abcam, Cambridge, MA) were added to the resulting supernatant and pre-clearing carried out at 4°C overnight. Following addition of mAb 61.3 (*Holder et al., 1985a*), samples were incubated at 4°C overnight before antigen-antibody complexes were precipitated using Protein G-Sepharose beads overnight at 4°C. The beads were washed five

times in wash buffer I (50 mM Tris-HCl pH 8.2, 5 mM EDTA, 0.5% Nonidet P-40, 1 mg/mL bovine serum albumin, 0.5 M NaCl) and twice in wash buffer II (50 mM Tris-HCl pH 8.2, 5 mM EDTA, 0.5% Nonidet P-40) before antigen-antibody complexes were eluted using NuPAGE LDS Sample Buffer (Life Technologies, Carlsbad, CA) and proteins resolved using precast NuPAGE Novex 3–8% Tris-Acetate protein gels (Life Technologies). Following electrophoresis, samples were evaluated by immunoblot analysis probing with anti-Ag44 or anti-RhopH1/Clag3.1 antibodies followed by horse-radish-peroxidase secondary antibodies as described (*Jean et al., 2003*).

## Immunofluorescence microscopy

Immunofluorescence microscopy was performed on formaldehyde-fixed thin blood smears, permeabilised with 0.1% (v/v) Triton X-100. Monoclonal anti-RAP2 (MRA-876), directly labeled with Alexa Fluor 594 using the Alexa Fluor 594 Antibody Labelling Kit (Life Technologies), was used at a dilution of 1:300. Samples were probed with primary antibodies used at the following dilutions: anti-Ag44 (1:2000), mAb 61.3 (1:100), anti-CL3.1A (1:100), anti-MAHRPI (1:2000), anti-KAHRP (1:250), rabbit anti-MSP1 (1:1000), mAb 89.1 (1:1000), and mAb 7.7 (1:100). Bound primary antibodies were detected using Alexa Fluor 488-, 566- or 594-conjugated anti-rabbit or anti-mouse secondary antibodies (Life Technologies), diluted 1:8000. Slides were mounted in ProLong Gold Antifade Mountant with DAPI (Life Technologies) and trophozoite images captured using a Nikon Eclipse Ni-E widefield microscope with a 100x/1.45NA objective and a Hamamatsu C11440 digital camera. Schizont stage images were captured with a Zeiss LSM 880 using a 63x/1.4 NA objective equipped with an Airyscan detector to improve the optical resolution of the scanned images. The DAPI, Alexa Fluor 488 and Alexa Fluor 594 channels were imaged sequentially over the axial dimension and processed using the integrated Zeiss software to enhance the optical resolution isometrically ~1.8 fold. All images were processed using either the Zen 2012 or FIJI software packages. For display purposes, linear adjustments were made to the intensity scale of each channel to equalize the intensity output to enhance areas of co-localization. Relative intensities between samples are not comparable.

## Transmission electron microscopy

Parasite cultures ~92 hr following rapamycin (or mock) treatment were fixed at 37°C in 8% formaldehyde in 0.2 M phosphate buffer pH 7.4 (PB) for 15 min by adding 1 vol of fixative solution to 1 vol of culture. The cells were pelleted, then further incubated in 2.5% glutaraldehyde, 4% formaldehyde in 0.1 M PB at room temperature for a further 30 min. Cells were washed in 0.1 M PB before being embedded in 4% (w/v) low-melting point agarose in distilled water. The agarose-embedded samples were cut into 1 mm$^3$ blocks, post-fixed in 1% (w/v) OsO$_4$ and 1.5% (w/v) potassium ferrocyanide for 60 min at 4°C then incubated sequentially in 1% (w/v) tannic acid in 0.05 M PB for 45 min and 1% (w/v) sodium sulphate in 0.05 M PB for 5 min. The samples were washed in water and dehydrated through a graded series of ethanol before being embedded in Epon resin (Taab 812). Blocks were trimmed and ultrathin 70 nm sections cut using a diamond knife on a UC6 Ultramicrotome (Leica Microsystems, Wetzlar, Germany), picked up on 150 hexagonal mesh copper grids and post stained with lead citrate before being imaged using a Tecnai G2 Spirit 120 kV transmission electron microscope (FEI Company, Hillsboro, OR) with an Orius camera (Gatan Inc., Pleasanton, CA).

## Analysis of erythrocyte membrane permeability

Sorbitol sensitivity of parasites was determined 72 hr following rapamycin or DMSO treatment (that is, in cycle 2). Cultures at equal parasitaemia were incubated in osmotic lysis buffer (280 mM sorbitol, 20 mM Na-HEPES, 0.1 mg/mL BSA, pH 7.4) for 7 min, then hemoglobin release determined by measuring the absorbance of the cell supernatants at 405 nm, as previously described (*Ginsburg et al., 1985*; *Kirk et al., 1994*).

5-ALA uptake was determined by incubating cultures of synchronous cycle 2 ring-stage parasites overnight in phenol red-free RPMI 1640 medium (K-D Medical, Columbia, MD) supplemented with 200 µM 5-ALA. Just prior to analysis, parasite nuclei were stained by treatment with 2 µM Hoechst. PPIX and Hoechst fluorescence were captured using a Zeiss LSM 880 equipped with a 63x/1.4 NA objective in standard confocal detection mode. Images were captured with the same acquisition setting so that measurements of intensity are directly comparable. Co-occurrence of PPIX and Hoechst was quantified using the MetaMorph software 'Cell Scoring' application. Cells were also analyzed on

a SORP LSRFortessa, detecting PPIX emission with a 532 nm laser through a 605/40 nm bandpass filter and Hoechst emission with the 406 nm laser through a 440/40 nm bandpass filter. Erythrocyte doublets were excluded using a FCS-A versus FCS-H display and data analyzed by BD FACSDiva software.

## Time-lapse video microscopy

*P. falciparum* egress was imaged as previously described (*Collins et al., 2013*; *Das et al., 2015*), using 1 μM (4-[7-[(dimethylamino)methyl]−2-(4-fluorphenyl)imidazo[1,2-$\alpha$]pyridine-3-yl]pyrimidin-2-amine (compound 2) to tightly synchronise egress. Following removal of compound 2 by washing, parasites were suspended in fresh pre-warmed medium and introduced into a pre-warmed microscopy chamber on a temperature controlled microscope stage at 37°C. Beginning 6 min after washing off the compound 2, DIC images were collected at 5 s intervals for 30 min using a Nikon Eclipse Ni Microscope fitted with a Hamamatsu C11440 digital camera and converted to QuickTime movies using Nikon NIS-Elements software.

## Acknowledgements

This work was supported by the Francis Crick Institute, which receives its core funding from Cancer Research UK (FC001043 and FC001097), the UK Medical Research Council (FC001043 and FC001097), and the Wellcome Trust (FC001043 and FC001097). ESS was supported by a Wellcome Trust/NIH PhD studentship (013459/Z/14/Z) and CvO was supported by a Wellcome Trust Career Re-Entry Fellowship (095836/Z/11/Z). This research was also supported in part by the Intramural Research Program of the National Institute of Allergy and Infectious Diseases, National Institutes of Health. The authors are indebted to Anne Weston of the Electron Microscopy Science Technology Platform at the Francis Crick Institute, Fiona Hackett (Francis Crick Institute) for assistance with parasite culture, J. J. Lopez-Rubio (University of Montpellier) for sharing the pL6 and pUF1 plasmids and David Stephany and Kevin Holmes (Flow Cytometry Core, Research and Technologies Branch, National Institute of Allergy and Infectious Diseases) for assistance with cell analysis. The following generously provided antibodies: R Coppel, O Kaneko, A Holder, J Vakonakis, BEI resources via contribution from A Saul and The European Malaria Reagent Repository via contribution from J McBride.

## Additional information

### Funding

| Funder | Grant reference number | Author |
| --- | --- | --- |
| Wellcome | 013459/Z/14/Z | Emma S Sherling |
| National Institutes of Health | | Emma S Sherling<br>Joseph A Brzostowski<br>Louis H Miller |
| Wellcome | FC001097 | Ellen Knuepfer |
| Cancer Research UK | FC001097 | Ellen Knuepfer |
| Medical Research Council | FC001097 | Ellen Knuepfer |
| Wellcome | FC001043 | Michael J Blackman |
| Cancer Research UK | FC001043 | Michael J Blackman |
| Medical Research Council | FC001043 | Michael J Blackman |
| Wellcome | 095836/Z/11/Z | Christiaan van Ooij |

The funders had no role in study design, data collection and interpretation, or the decision to submit the work for publication.

## Author contributions

ESS, Conceptualisation, Formal analysis, Funding acquisition, Conceived and designed the experiments, Investigation—performed the experiments and analysed the data, Writing—original draft, Writing—review and editing; EK, Investigation—designed the Southern blotting strategy; JAB, Investigation—assisted with image acquisition; LHM, Supervision, Funding acquisition; MJB, CvO, Conceptualisation, Supervision, Funding acquisition, Conceived and designed the experiments, Writing—original draft, Writing—review and editing

## Author ORCIDs

Emma S Sherling, http://orcid.org/0000-0002-8339-7060
Michael J Blackman, http://orcid.org/0000-0002-7442-3810
Christiaan van Ooij, http://orcid.org/0000-0002-6099-6183

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
