## [Decision Letter]

Thank you for submitting your article "The *Plasmodium falciparum* rhoptry protein RhopH3 plays essential roles in host cell invasion and nutrient uptake" for consideration by *eLife*. Your article has been favorably evaluated by Michael Marletta (Senior Editor) and three reviewers, one of whom, Dominique Soldati-Favre (Reviewer #1), is a member of our Board of Reviewing Editors.

The reviewers have discussed the reviews with one another and the Reviewing Editor has drafted this decision to help you prepare a revised submission.

Summary:

*Plasmodium falciparum* modifies host erythrocytes to render them permeable to nutrient uptake. The mechanisms underlying formation of plasmodial surface anion channel /permeability pathways (PSAC/NPP) are not fully understood. Using an elegant inducible Cre-lox approach to truncate RhopH3, Sherling et al. show that RhopH3 plays a critical role during *Plasmodium falciparum* growth inside erythrocytes, and provide data supporting a role of the RhopH complex in the function of NPPs.

First, the authors showed that truncation of RhopH3 led to mis-targeting of other members of the RhopH complex but that RhopH1 and RhopH2 can still interact with each other. These parasites mutants were not defective in egress but partially affected in their ability to invade new host cells. Examination of these mutants in the second cycle of infection clearly uncovered a block in intraerythrocytic development. Importantly, the export of parasite proteins into the host cell and Maurer's cleft formation were not altered in the erythrocytes infected with parasites lacking full-length RhopH3 however they became resistant to sorbitol lysis, consistent with a lack of NPP activity, and showed an impairment in the update of 5-aminolevulinc acid.

Essential revisions:

The manuscript is clearly written. The study is very well performed, and provides novel and important insights into the function of a critical protein complex involved in *Plasmodium falciparum* growth inside erythrocytes. However, some issues have been raised by the three reviewers and in order to improve the manuscript, the authors should address a number of points listed below. In particular, the study provides genetic evidence supporting a role of the RhopH complex in NPP function, but evidence supporting a role during invasion is much weaker.

1) The authors claim that RhopH3 mutants are impaired during invasion. However, there is not sufficient direct evidence to support this conclusion. Invasion was analyzed 48 hours after addition of fresh erythrocytes to rapamycin-treated trophozoites, thus during cycle 2. The observed decrease in infection may be a consequence of growth arrest or parasite death (due to nutrient depletion) during the 48h window rather than reduced invasion. The assay used to assess invasion efficiency is a nice one but it does not discriminate with an indirect invasion defect. The authors should use live cell microscopy (as they did to investigate egress) to look at the ability of the egressed parasites to invade new erythrocytes.

2) The floxed parasites need to be better characterized. Are the levels of RhopH3 (and other Rhop) proteins in floxed parasites normal in comparison to parental WT? Do these parasites grow normally in the absence of rapamycin treatment?

Also, the fate of RhopH components, including RhopH3 itself full length versus truncated over the course of rapamycin treatment should be documented for correct interpretation of the phenotypic assays. In particular, at end of cycle 1 (egress/invasion assays) or cycle 2 (electron microscopy), do parasites express truncated RhopH3 only? Is truncated RhopH3 (and other component of RhopH) complex detectable in extracellular merozoites after treatment with rapamycin?

3) The wording around the defect in nutrient import (in the Abstract and in the relevant Results heading ("Import pathways are not functional…") is too strong and too general, given that only sorbitol entry and 5-ALA entry were examined, using indirect methods. It would be relevant to know if RhopH3Δ4-6 are defective in the uptake of isoleucine. Measuring the uptake of radiolabelled isoleucine by erythrocytes infected with the different parasite lines would be ideal and would certainly strengthen the paper. Alternatively, the authors should use more cautious wording (e.g. that their results are consistent with a defect in the activity of the NPPs).

---

## [Author Response]

*Essential revisions:*

*The manuscript is clearly written. The study is very well performed, and provides novel and important insights into the function of a critical protein complex involved in Plasmodium falciparum growth inside erythrocytes. However, some issues have been raised by the three reviewers and in order to improve the manuscript, the authors should address a number of points listed below. In particular, the study provides genetic evidence supporting a role of the RhopH complex in NPP function, but evidence supporting a role during invasion is much weaker.*

*1) The authors claim that RhopH3 mutants are impaired during invasion. However, there is not sufficient direct evidence to support this conclusion. Invasion was analyzed 48 hours after addition of fresh erythrocytes to rapamycin-treated trophozoites, thus during cycle 2. The observed decrease in infection may be a consequence of growth arrest or parasite death (due to nutrient depletion) during the 48h window rather than reduced invasion. The assay used to assess invasion efficiency is a nice one but it does not discriminate with an indirect invasion defect. The authors should use live cell microscopy (as they did to investigate egress) to look at the ability of the egressed parasites to invade new erythrocytes.*

With the benefit of hindsight, we agree that the assay as originally performed to examine invasion efficiency in the RhopH3 mutant parasites (Figure 3 in the original manuscript) was not optimal for the reasons given above. This issue has now been redressed in the revised manuscript (new Figure 3) by including a more ‘standard’ invasion assay, in which mature synchronous schizonts of rapamycin-treated or control RhopH3 mutants were purified, added to fresh red blood cells, then parasitaemia levels of newly invaded ring stage parasites determined just 4 h later. This method measures the efficiency of new ring formation over a very short period, and is therefore much more suitable for determining invasion efficiency, independent of the capacity of the intraerythrocytic parasites to subsequently develop. Note that these revisions have also required some minor textual changes to the main text (subsection “Loss of the RhopH complex leads to an invasion defect”, last paragraph) and to the Figure 3 legend.

We respectfully suggest that the reviewers’ suggestion to use live cell microscopy to determine the efficiency of invasion is unsuitable for this purpose. We already know that the parasites form rings at the beginning of cycle 2, proving that some of them (~50%) do successfully invade. However, the technical challenges of imaging invasion in real time means that very few individual invasion events can be monitored per egress event (see for example Gilson and Crabb Int J Parasitol 39:91-96, 2009), with the result that it is not a suitable approach for *quantifying* invasion efficiency.

*2) The floxed parasites need to be better characterized. Are the levels of RhopH3 (and other Rhop) proteins in floxed parasites normal in comparison to parental WT? Do these parasites grow normally in the absence of rapamycin treatment?*

We accept that this issue was not formally addressed in the original manuscript. The revised manuscript now contains an additional supplementary figure (new Figure 1—figure supplement 2) showing that; (1) the non-rapamycin-treated 5F5 and 4B11 parasites produce wild type levels of RhopH3 and RhopH1; and (2) that they grow normally, as compared to the parental 1G5DC parasites. This confirms that floxing the *rhopH3* gene did not detectably affect RhopH3 expression, nor impact on parasite viability.

*Also, the fate of RhopH components, including RhopH3 itself full length versus truncated over the course of rapamycin treatment should be documented for correct interpretation of the phenotypic assays. In particular, at end of cycle 1 (egress/invasion assays) or cycle 2 (electron microscopy), do parasites express truncated RhopH3 only?*

We realize that in our original manuscript, the legend to Figure 1 was not explicit enough. This text has now been amended to make it clear that only truncated RhopH3 was detectable by the end of cycle 1 in schizonts of rapamycin-treated parasites. Since the cycle 2 parasites were all derived from the cycle 1 schizonts, we assume that the cycle 2 parasites also expressed exclusively truncated RhopH3. Note that this could not be formally proven by Western blot since the cycle 2 parasites never reached schizont stage (at which RhopH3 levels are maximal); however, the prominent phenotype displayed by the cycle 2 parasites is consistent with them expressing the mutant RhopH3 as expected.

*Is truncated RhopH3 (and other component of RhopH) complex detectable in extracellular merozoites after treatment with rapamycin?*

We appreciate that this is a pertinent question, since the mislocalisation induced by truncation of RhopH3 could potentially result in merozoites that entirely lack the RhopH proteins. To address this, we have now performed immunofluorescence analysis of naturally egressed merozoites of the RhopH3 mutants. The results, showing mislocalisation and loss of RhopH3 from many free merozoites, have now been appended to Figure 2 (new Figure 2) to produce a modified form of that figure in the revised manuscript.

*3) The wording around the defect in nutrient import (in the Abstract and in the relevant Results heading ("Import pathways are not functional…") is too strong and too general, given that only sorbitol entry and 5-ALA entry were examined, using indirect methods. It would be relevant to know if RhopH3Δ4-6 are defective in the uptake of isoleucine. Measuring the uptake of radiolabelled isoleucine by erythrocytes infected with the different parasite lines would be ideal and would certainly strengthen the paper. Alternatively, the authors should use more cautious wording (e.g. that their results are consistent with a defect in the activity of the NPPs).*

We entirely accept the reviewers’ point, and have now modified the text of this subtitle to: ‘Import pathways are defective in rhopH3 mutant parasites’. We trust that this is acceptable.